# Healthcare Liability and Thyroidectomy: When Is the Surgeon Wrong?

**DOI:** 10.3390/healthcare11040577

**Published:** 2023-02-15

**Authors:** Maricla Marrone, Carlo Angeletti, Mirko Leonardelli, Stefano Duma, Gerardo Cazzato, Ettore Gorini, Alessandro Dell’Erba, Cristoforo Pomara

**Affiliations:** 1Section of Legal Medicine, Department of Interdisciplinary Medicine, Bari Policlinico Hospital, University of Bari, 70124 Bari, Italy; 2Section of Molecular Pathology, Department of Emergency and Organ Transplantation (DETO), University of Bari “Aldo Moro”, 70124 Bari, Italy; 3Department of Economics and Finance, University of Bari “Aldo Moro”, 70124 Bari, Italy; 4Department of Medical, Surgical and Advanced Technologies “G.F. Ingrassia”, University of Catania, 95121 Catania, Italy

**Keywords:** thyroid surgeries, medical malpractice, complications, Italian legislation

## Abstract

Thyroid surgeries can often lead to operative complications, sometimes with consequences on the patient’s health. This often leads to claims for compensation but the assessments of consultants and judges are not always objective. Based on these considerations, the authors analyzed forty-seven sentences issued between 2013 and 2022 regarding claims of alleged medical malpractice. This analysis aims to examine the cases presented in the sentences and the evaluations proposed by the judges to offer ideas for objective evaluation by the legislation in force in Italy.

## 1. Introduction

In medicine, when can we speak of a “complication”, and when is this “complication” not attributable to a medical error, i.e., when is it “excusable”?

In Italy, the Supreme Court of Cassation (n. 13328, 30 June 2015) stated that “With the term ‘complication’ clinical medicine and forensic medicine usually designate a harmful event that although abstractly foreseeable, would not be avoidable. The law is not concerned with whether or not the harmful event not intended by the physician falls within the clinical classification of complications: it is only concerned with whether that event integrates the extremes of ‘unattributable cause’. The circumstance that an unwanted event is qualified by the clinic as a “complication” is not enough to make it per se a “nonattributable cause” within the meaning of Article 1218 of the Civil Code; just as, on the other hand, events that do not qualify as complications can theoretically constitute fortuitous events that exclude the physician’s fault. It follows from the foregoing, on the level of proof, that in the judgment of liability between patient and physician: -either the physician succeeds in proving that he has behaved in accordance with the leges artis, and then he is exempt from liability regardless of whether or not the harm suffered by the patient falls into the category of “complications”; -or, on the contrary, the physician fails to provide that proof: and then the circumstance that the harm event is in the abstract unforeseeable and unavoidable will not benefit him, since what matters is whether it was foreseeable and avoidable in the concrete case. Foreseeability and avoidability in the concrete case, which, insofar as said, it is the physician’s burden to prove.” (Court of Cassation, Sec. III Civil, 30 June 2015, No. 13328).

In the area of health care liability, there are clearly areas where complications occur more frequently and are therefore more often the subject of court litigation.

Thyroid surgeries are frequently the subject of forensic doctors’ evaluations if the patient complains of a worsened state of health as a result of them.

The thyroid gland is an endocrine gland located in the anterior part of the neck, in front of the trachea. Structurally, it consists of two lobes joined on the midline by an isthmus, giving it a characteristic “butterfly” appearance.

The thyroid gland is in close relationship with the laryngeal nerves that regulate the functioning of the vocal cords. In addition, arranged on its surface are the parathyroids, which receive blood from the same vessels that supply the thyroid gland. The surgeon, during thyroidectomy surgery—total or partial—must pay attention to these structures since proximity with the aforementioned organs can lead to the occurrence of complications.

The main complications that can be observed after thyroid surgery are RLN (recurrent laryngeal nerve) lesions in the form of bilateral or unilateral and temporary or permanent paralysis and temporary or permanent hypoparathyroidism.

Postoperative RLN injury is a relatively rare complication of thyroid surgery, the rate of permanent damage occurring approximately between 0.3 and 3% of cases, with transient paralysis occurring in up to 8% of cases. Paralysis is generally regarded as transient if the patient has recovered up to 12 months after surgery, and it is regarded as permanent if the paralysis remains unmodified 12 months after surgery.

Clinically, nerve damage results in clinical pictures of varying severity, depending on the branch injured. Isolated SLN (superior laryngeal nerve) lesions (those without involvement of the recurrent laryngeal nerve) result, in most cases, in clinical pictures with nuanced symptoms due to reduced tension in the vocal cord affected by the paralysis, resulting in a difficulty in modulating the voice for high-pitched tones. Reduced sensitivity of the involved hemilarynx may also be present with moderate difficulty in swallowing fluids.

In contrast, lesions of the RLN have important clinical significance, as RLN paralysis (recurrent paralysis) is manifested by the fixation of the vocal cord of the involved side. The most prominent symptom is dysphonia, which is of a greater magnitude the more the affected vocal cord is atrophic, flaccid, and far from the midline. In some cases, swallowing disorders with the occasional inhalation of fluids and coughing episodes during feeding are also present. Despite the severity of the injuries, however, most damage to the RLN is transient. Of different clinical significance is bilateral vocal cord paralysis—a serious clinical condition manifested by aphonia, respiratory obstruction, and difficulty swallowing.

The incidence of RLN lesions following thyroid surgery ranges from an unlikely 0%, as has been reported in older studies [1], to 8% [2], while SLN lesions are quite frequent (about 10%), although they are often underestimated and thus underreported [3]. The prevalence of lesions on the RLN increases significantly with thyroid reintervention [4]. In the Italian setting, a multicenter study reported that after total thyroidectomy, 3.7% of cases showed monoplegia (2.4% transient and 1.3% definitive) and 0.6% showed diplegia, half of which were treated with tracheotomies [5]. The same study found that permanent RLN injury occurred in 2% of patients, despite the fact that the surgeon had technically uncovered the nerve, carefully following it until it entered under the constrictor muscle of the pharynx, that is, without detecting any intraoperative problems indicative of damage to its anatomical integrity [5]. The legal implications are considerable, as more than half of the claims following thyroid surgery involve injuries to the RLN [6]. 

With the intention of reducing nerve injury during surgical dissection, after pioneering attempts, the method of intraoperative nerve monitoring (IONM) of the RLN has been introduced, and increasingly widely used, for thyroid and parathyroid surgery for more than two decades [7,8].

More recently, in accordance with the evolution that has taken place in the field of IONM, continuous intraoperative neural monitoring (CIONM) of the vagus nerve has been introduced [9], which offers the possibility of recording even small and initial changes in nerve transmission indicative of possible nerve distress. In reductive terms, variations in nerve signal allow surgical maneuvers that may potentially allow nerve damage to be modified during dissection. Although visual nerve identification proves to be the preferred technique for many surgeons to avoid nerve damage, the possibility of identifying nerve damage in the absence of macroscopically visible lesions, secondary to surgical manipulation, and thus to edema and paralysis of the RLN, is described. These injuries result mainly in transient nerve damage but can also result in permanent damage. Recent studies show that using neuromuscular monitoring results in a reduced incidence of transient RLN paralysis compared with visual identification alone [10,11,12,13,14,15,16,17,18,19].

Based on these considerations, the authors analyzed fifty sentences issued between 2013 and 2022 regarding claims for alleged medical malpractice.

## 2. Aim and Scope

This article aims to assess the nature of compensation claims for alleged medical malpractice in thyroid surgery in Italy. 

The authors aim to analyze the nature of the requests for compensation and the judges’ assessments proposed in the sentence in the context of alleged medical malpractice during thyroid surgery in Italy. In particular, also considering the new Italian law concerning health liability (Law 24/2017), the authors concentrated their assessment on:-the most frequently reported alleged damages;-any reprehensible conduct of the healthcare professionals;-the reasons for accepting or rejecting the compensation claims;-the extent of the compensations awarded (in terms of temporary total/partial disability and biological damage).

### 2.1. Materials and Methods

A retrospective study was conducted using the Pluris Wolters Kluwer legal database as well as the Telematics Services Portal (TSP) for searching judgments. The keywords used to select the judgments were medical-legal litigation, malpractice, medical-legal complaint, informed consent, guidelines, technology, and expertise, which were matched with each of the following terms: thyroid, thyroidectomy, hyperparathyroidism, parathyroidectomy, cervical lymph node metastasis, and cervical lymphadenectomy. 

We randomly selected fifty judgments issued from 2013 to 2022 in Italy concerning claims following thyroid surgery.

The aspects concerning the outcomes of the judges’ decisions were analyzed for each judgment in detail.

The Wolters Kluwer Pluris database is a continuously updated search engine that, upon subscription, allows searches for sentences and maxims, authorial commentaries to articles in major codes, scholarly articles, procedural assistance, and full texts relevant to the search. 

The Telematic Services Portal (TSP) of the Ministry of Justice, on the other hand, is a tool that allows the searching and viewing of merit judgments for REGINDE members only, without the need for subscription.

As part of the inclusion criteria, the completeness of each judgment was assessed with regard to the following data:the court’s jurisdiction and the date of judgment;the sex and age of the plaintiff/appellant;the underlying pathology;the type of surgery undergone;the alleged harm complained of;the outcome of the dispute;the damages awarded (if the claim was successful);the grounds for the judgment.

Of the fifty judgments analyzed, only forty-seven had the above criteria, while three were excluded from the statistical evaluation because they lacked data on the percentage value of compensation as well as details regarding the patients’ underlying diseases. 

Table 1 shows these data.

However, partly because of Italian privacy legislation, which is strict (in favor of protecting the rights of individuals), only forty-seven contained much of the data useful for the present study, although, even in the forty-seven judgments analyzed, some data about the plaintiff/appellant could not be found. 

For all the judgments, the outcome and the reason for each were made clear. This is crucial as it allows us to trace some recurring features.

### 2.2. Statistic Analysis

A statistical analysis was conducted using the Microsoft Excel 2013 software (Microsoft Corporation, Redmond, WA, USA) and IBM SPSS Statistics version 25 for Windows (IBM Corporation, Armonk, NY, USA). The categories examined were then represented in percentage terms.

## 3. Results 

Fifty judgments issued from 2013 to 2022 related to thyroid surgery were examined. Of these, forty-seven were complete with the necessary data for the following analysis. Forty-three out of forty-seven cases involved total thyroidectomy surgery, suffered by female individuals in forty cases (85.11%) and by male individuals in seven cases (14.89%). In twenty-seven of the cases (57.45%), the disorder complained of was due to recurrent nerve injury. In two cases in addition to recurrent nerve injury, the claim for compensation was based on alleged overtreatment, while in only one case, the claim was exclusively about overtreatment in the absence of nerve injury.

In two cases (4.26%), the injury complained of was due to a post-surgical hemorrhage. In one of these, in addition to post-surgical hemorrhage, there was a complaint of damage from improper tracheostomy packing with tracheal injury, post-anoxic encephalopathy, and spastic tetraparesis. In another case, the plaintiff complained of laryngospasm as a result of remaining thyroid tissue and subsequent lymphadenopathy.

In four cases, harm was caused by hypocalcemia associated with reactive anxiety disorder (in two cases), glottic stenosis, and RLN injury (in one case), and tetanic seizures (in an additional case). In addition, in one case, the injury complained of was due to post-surgical laryngeal edema with a respiratory crisis, cardiovascular arrest, and subsequent tracheostomy, while in an additional case, the injury was a result of septic shock and subsequent death.

Of the forty-seven rulings, thirty-four resulted in an upholding of the plaintiff/appellant’s claim (72.34 percent) and thirteen rejected it (27.66 percent).

The judgments that provided an award for damages, issued since 2018, have provided for liquidated damage, in percentage terms, bounded in a range between 8 and 40 percent, with an average value of 21.53 percent. There are, in fact, no percentage values of damage awards below 8 percent among the judgments analyzed.

In contrast, the judgments pertaining to previous facts were evaluated in a range between 3 and 22 percent with an average value of 13.4 percent, with the exception of only one judgment (No. 41, Table 1) in which the patient was awarded with 90 percent biological damage as a result of improper tracheostomy packing performed by the surgeon in the absence of the anesthesiologist, resulting in post-anoxic brain damage and spastic tetraparesis. (Figure 1).

## 4. Discussion

According to a recent literature search, the recurrent laryngeal nerve is the most injured in thyroid surgery. Lesions of the recurrent laryngeal nerve can lead to various serious clinical situations. According to several studies, the thyroid surgeries most involved in nerve damage are for recurrent goiter, thyroid tumors, and Grave’s disease. To avoid damage to the anatomical structures close to the thyroid, it is important to know the anatomy of the anatomical region of the neck well. In addition to the presence of nerves, there are many vessels which, if damaged, could lead to dangerous complications. For example, according to a recent meta-analysis, it is important to know during thyroid surgery that approximately 3.8% of individuals have a variant of the thyroid artery (TIA). This awareness on the part of the surgeon can prevent bleeding. [20,21].

In most of the sentences examined by the authors (72.34 per cent), there was an acceptance of the plaintiff/appellant request.

The decisions to grant the claims were all motivated on the basis of deficiencies in the plaintiffs’ medical records, particularly with regard to the descriptions of the surgical procedures performed: the absence of detailed descriptions on how the identification and preparation of the laryngeal nerves operated on were conducted and of the technique performed, was a source of the attribution of liability. The lack of an operational accuracy description favored the iatrogenic injury hypothesis.

In Italy, “the medical record prepared by the doctor of a public health facility has the nature of a public act endowed with privileged faith with reference to the facts attested by it” (Court of Cassation Criminal Section V, Judgment 11 September 2013 No. 37314). At the same time, “the incompleteness of the medical records is a factual circumstance that the judge can use to deem proven the existence of a valid causal link between the doctor’s actions and the damage suffered by the patient. If there are no clinical documentations that exclude the doctor’s error, it can be assumed that the doctor caused the damage” (Court of Cassation Civil Section III, Judgment 14 November 2019 no. 29498).

In our legal system, if the patient proves that he has suffered damage after medical treatment, the doctor or hospital has the burden of proving that they are not responsible.

In thirty-four of the cases examined, which ended favorably for the injured party, the sanitarians were unable to prove that the services performed were properly performed and that the non-performance was due to a cause not attributable to them.

Instead, it was proven by the injured party that the conduct of the health care providers was likely to lead to the event of injury, in line with previous rulings of legitimacy (Court of Cassation, Civil Sect. III, ruling 5–26 July 2017, no. 18392; Court of Cassation, Civil Sect. III Civil, judgment 14 November 2017, No. 26824; Court of Cassation, Sec. III Civil 7 December 2017, No. 29315; Court of Cassation, Sec. III Civil, 15 February 2018, No. 3704; Court of Cassation, Sec. III Civil, 23 October 2018, No. 26700; Court of Cassation, Sec. III Civil, 11 November 2019, No. 28991).

In the remaining cases, the judges’ aides assessed the conduct of the patients as correct after assessing the correctness and completeness of the medical records.

In one of the cases reviewed, liability was not attributed to the medical professionals because of the particular difficulty of the specific case (Judgment 1, Table 1). In fact, the thyroid gland was of such a size that the anatomical relationships of the gland were altered. Therefore, the cause of the damages complained of (dysphonia and dysphagia) was attributed to the injury of the plaintiff’s nerve, which was not detected during the operation, and the claim for compensation was not upheld, since the sanitarians, under the conditions in which they were operating, could not prevent what in fact happened. In fact, in Italy, under Article 2236 of the Civil Code, “If the service involves the solution of technical problems of special difficulty, the service provider is not liable for damages, except in cases of malice or gross negligence.” In our country, in the field of health care liability, this rule does not apply to the case of damages attributable to the negligence and imprudence of the professional, but is limited to cases of inexperience attributable to the special difficulty of technical problems that the professional activity, in concrete terms, must deal with (Court of Cassation, Sec. III Civil, 19 April 2006, No. 9085), with the clarification that, according to the principle of closeness of proof, it is up to the doctor to prove the particular difficulty that characterized the case (Court of Cassation, Sec. III Civil, 9 November 2006, No. 23918).

In fact, let us briefly recall that the recurrent nerve (or inferior laryngeal nerve) consists of somatomotor, somatosensory, and visceral fibers. It has sphincteric (protecting the airway from foreign bodies), respiratory, and phonatory functions [22]. Even if, during its course, the recurrent nerve contracts relationships which, due to their complexity, can cause surgical problems, careful isolation (of the nerve) can significantly reduce the risk of complications [23,24,25].

Recurrent nerves often suffer ischemic or mechanical damage [26]. In the case evaluated, the health professionals were found not guilty as the recognition and isolation of the recurrent nerve was particularly difficult due to the pathological anatomical alterations of the thyroid gland. Often, however, the assessment of the judge’s consultant, and therefore of the judge, in these cases, varies based on subjective criteria as the line between avoidable error and complication is thin.

Moreover, even following the introduction of new intraoperative nerve monitoring technologies (IONM), of all the judgments analyzed, as far as can be inferred from a careful reading of them, in no case had such monitoring systems been used. This, it is assumed, is in relation to the high costs inherent in the purchase of such equipment [25]. Here, then, is the question of how far the surgeon, although having performed the surgery correctly, could have achieved the early recognition of the nerve injury in the absence of macroscopic alterations. Above all, even in light of L. 24/2017, and in accordance with L. 81/08, which accounts for the provision by the management of the appropriate instruments for the work activity of the surgeon, in none of the judgments analyzed was the responsibility of the health management highlighted, implying the failure to provide such instrumentation [27].

Further evaluation of the data on the rulings issued after Law 24/2017 is needed regarding the establishment of the Medical College [28]. Law 24 of 2017, in fact, after describing the safety of care and health risk management, the profiles of criminal and civil liability of health professionals, and the insurance obligations of the latter and of public and private health agencies, describes in Article 15 that “the judicial authority entrust the performance of technical advice and expertise to a doctor specializing in forensic medicine and one or more specialists in the discipline who have specific and proven knowledge of what is the subject of the proceedings.” In accordance with the legislature, the Superior Council of the Judiciary, on 25 October 2017, through a recommendation of the VII Commission, deemed “new action appropriate with regard to the appointments by judicial authorities in both the criminal and civil sectors to all auxiliaries to be appointed in proceedings specifically dealing with medical liability.”

Thus, since 2017, following Law 24/2017, if disputes regarding health liability in Italy are evaluated by a forensic doctor who must be supported by a specialist in the specific sector with “proven experience”, and thus are allowed a more objective and precise evaluation, everything that was previously judged was often devoid of this double evaluation, which is necessary. The expert surgeon can, in fact, on the basis of the data available, evaluate whether a complication is unavoidable or whether it is a mistake. The associated expert in forensic medicine can allow the boundaries of these terms (complication/error) to be better defined, avoiding clinical confusions, which cannot be transposed to the legal context.

Moreover, the forensic doctor will be able to correctly quantify the patient’s damage. In this regard, in fact, one of the data emerging from the analysis of the judgments concerns the assessment of biological damage formulated by the consultants before and after the introduction of the legal device and therefore before and after a collegial evaluation.

As has already been said, the judgments that have provided an award for damages issued since 2018 have provided for liquidated damage, in percentage terms, bounded in a range between 8 and 40 percent, with an average value of 21.53 percent. 

In contrast, the judgments pertaining to previous facts were evaluated in a range between 3 and 22 percent with an average value of 13.4 percent.

This result, in the opinion of the authors, allows us to state how, as a result of the evaluation carried out by the medical examiner and the branch specialist, it was possible to obtain a more homogeneous range of evaluations with recognition, both in absolute terms and relative to individual cases, of decidedly higher percentages. Here, then, is a confirmation of the importance and indispensability of dual evaluations involving a forensic doctor and specialist, in cases of medical liability.

Nonetheless, however, when faced with a standard thyroid surgery (not burdened with a particular technical difficulty) that produces recurrent nerve damage, evaluators are often asked to indicate which and how many elements should be considered and with what degree of importance.

The criteria commonly used in the judgments analyzed concerned the correct identification of the nerve in association with its adequate isolation during surgery (documented in the operative report). Assessments were was also made on whether specialist postoperative evaluations of the patients had been performed. Another parameter evaluated almost homogeneously in the judgments concerns the timing of the onset of the symptomatology complained of by the plaintiffs, which was considered by almost all the judges (both judges and CTUs) as fundamental to the recognition of a debeatur.

An important finding from this study allows us to note that, in all the judgments analyzed, whenever the injury complained of was due to a complete injury of the LRN, mono or bilateral, the outcome of the evaluation saw the claim granted, while, in cases of partial injury or stretching, in about half of the cases this outcome was considered as a complication and not an error on the part of the health care providers.

The authors wonder, however, if it is not possible to also analyze other elements of the clinical documentation and if the judgment expressed can be considered objective or tainted by subjectivity.

Too often, requests for compensation following surgery for thyroid pathologies have different evaluation outcomes depending on the judge (or consultant). This favors the evident difference in attitudes towards the operating healthcare workers, which also causes difficult management of the medical work. If it is often not easy to make the clinical distinction between a foreseeable but not preventable complication (therefore one that is not attributable to the healthcare professional) and an error, it is certainly easier as well as more “objective” to make this distinction based on predominantly juridical criteria.

It is certainly easier to be able to judge if there are clear data in the medical records and if the operating report is written correctly.

The surgical expert in thyroid pathology who must evaluate the documentation can therefore more easily evade responsibility, thanks to a well-made medical record, even if the patient has suffered damage.

Furthermore, a good forensic doctor must perform the evaluation and must have experience in the percentage evaluation of damages to avoid discrepancies in the compensation to the patient.

The analysis of the judgments proposed by the authors has shown that there are also important differences in the quantification of the recognized damage: the range is very wide, ranging from 8 to 35 percent (if we exclude a single sentence in which the damage reported had caused spastic tetraparesis estimated as 90 percent of biological damage).

The analysis of the sentences did not allow the authors to understand how the each of the judges’ consultants arrived at proposing the specific percentage of damage. It was therefore not possible to explain these differences in detail, even if they aare almost always the consequence of subjective evaluations.

## 5. Conclusions

In Italy, the law, on the one hand, requires that the rating of cases of health liability is based only on documented evidence, but on the other hand, it depends on the subjective judgment of the auxiliary of the judge. The risk that this form of assessment could lead to wrong judgments made it necessary to amend the existing legislation. 

Law n. 24 of 2017 (already cited above) thus defined more clearly both the legal procedures within the process (of health responsibility) and the necessary characteristics that the consultants of the judge must have. The new law has also protected individual doctors in cases where the error was an organizational consequence and not a personal one.

All these elements introduced by the new law, together with the already known scientific knowledge, must be the basis both for the selection of consultants by the judge and for the evaluation of the consultants chosen. The “experience” of the judge’s consultants is certainly an important element in guaranteeing the correctness of the judgement. If, in addition to this, it were possible to trace a surgeon’s activity when he operates or to be able to demonstrate through suitable instruments the surgical activity step by step, perhaps evaluative homogeneity could be more easily achieved. In the meantime, having the health documentation correctly compiled and kept is a sure way to aid the doctor’s defense. Furthermore, it would be useful for forensic doctors and experts to periodically meet to arrive at a shared evaluation guideline that specifies which complications are unavoidable and which are errors.

All of this would be desirable in standardizing justice in terms of health liability and therefore in terms of compensation for damages for patients, but also to avoid unnecessary lawsuits and allow health professionals to operate more serenely.

## Figures and Tables

**Figure 1 healthcare-11-00577-f001:**
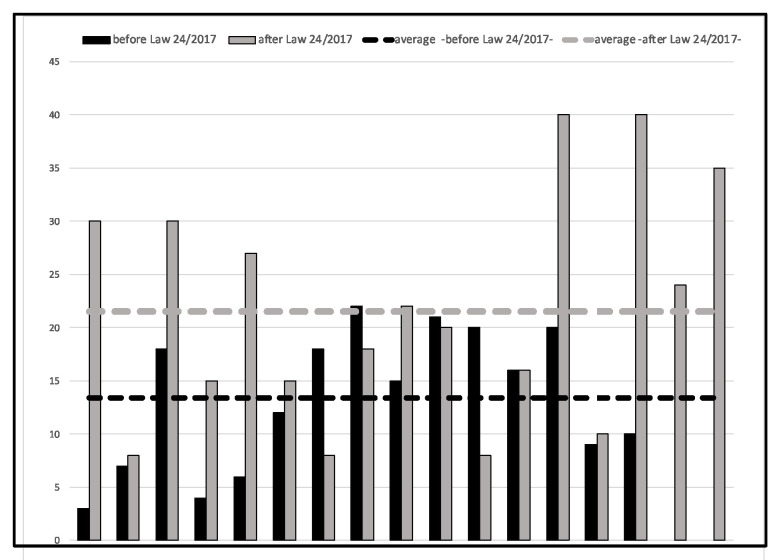
Assessment of biological damage before/after Law 24/2017.

**Table 1 healthcare-11-00577-t001:** (characteristics analyzed) (N.D.: not defined).

	Court	Date of Judgment	Gender/Age of Appellant	Disease	Type of Surgery	Alleged Damage	Outcome of the Judgment	Compensation Awarded	Grounds of the Judgment
1	Court of Velletri	March 2022	F-73	Suspected carcinoma	Total thyroidectomy	Overtreatment/transient paralysis RLN	Rejected	None	Complication
2	Court of Sassari	January 2022	F-N.D.	Goiter	Total thyroidectomy	Overtreatment/bilateral definitive paralysis RLN	Accepted	Biological damage: 30% Total disability: 0 Partial disability at 75%: 60 days Partial disability at 50%: 60 days	Bad identification/overstretching of the laryngeal nerves
3	Court of Milan	January 2022	F-42	Recurrence of thyroid cancer	Total thyroidectomy	Post-surgical bleeding	Accepted	Biological damage: 8% Total disability: 30 days Partial disability at 75%: 30 days Partial disability at 50%: 30 days Disability at 25%: 90 days Disability at 15%: 2 years	Incorrect removal of drains
4	Court of Pisa	October 2021	F-69	Goiter	Total thyroidectomy	Definitive paralysis right RLN and transient paralysis left RLN	Rejected	None	Complication
5	Court of Naples	May 2021	F-55	Goiter	Total thyroidectomy	Laryngeal spasms from permanence of thyroid residues	Rejected	None	Complication
6	Court of Rovigo	April 2021	F-58	N.D.	Total thyroidectomy	bilateral definitive paralysis RLN	Accepted	Biological damage: 30% Partial disability at 75%: 30 days Partial disability at 50%: 30 days	Failure to prepare laryngeal nerve
7	Court of Novara	March 2021	F-68	Chronic thyroiditis and Goiter	Total thyroidectomy	bilateral definitive paralysis RLN	Rejected	None	Complication
8	Court of Cosenza	February 2021	F-45	N.D.	Total thyroidectomy	Right definitive paralysis RLN	Accepted	Biological damage: 15%	Bad identification of the laryngeal nerves
9	Court of Rimini	February 2021	F-49	Goiter	Total thyroidectomy	bilateral definitive paralysis RLN	Accepted	Biological damage: 27% Total disability: 42 days Partial disability at 50%: 60 days Disability at 25%: 90 days	Bad identification of the laryngeal nerves
23	Court of Palermo	January 2021	M-32	Chronic thyroid disease	Total thyroidectomy	Hypothyroidism	Accepted	Biological damage: 15%	Incomplete medical record
10	Court of Catania	November 2020	F-71	N.D.	Total thyroidectomy	Left definitive paralysis RLN	Accepted	Biological damage: 8%	Lack of post-operation laryngeal nerve integrity check
11	Court of Pisa	November 2020	F-N.D.	Discovery of thyroid cancer cells on ovarian stroma	Total thyroidectomy	Overtreatment	Accepted	Biological damage: 18% Total disability: 10 days Partial disability at 50%: 20 days Disability at 25%: 30 days	Lack of indication to surgery
12	Court of Milan	August 2020	M-27	M. Basedow	Right partial thyroidectomy and subsequent left partial thyroidectomy	Hypercalcaemia and bilateral definitive paralysis RLN	Accepted	Biological damage: 22% (differential damage from 40% to 18%) Total disability: 10 days Partial disability at 50%: 20 days Disability at 25%: 30 days	Lack of post-operation right laryngeal nerve integrity check before performing the second surgery
13	Court of Palermo	August 2020	F-65	N.D.	Total thyroidectomy	Bilateral definitive paralysis RLN	Accepted	Biological damage: 20% Total disability: 10 days Partial disability at 50%: 60 days	Lack of post-operation laryngeal nerve integrity check
14	Court of Catania	May 2020	F-41	Goiter	Total thyroidectomy	Dysphonia, dysfagia and dyspnoea in transient bilateral paralysis RLN	Rejected	None	Complication
15	Court of Lecce	March 2020	F-61	Carcinoma	Total thyroidectomy	Injury of the right laryngeal nerve	Accepted	Biological damage: 8% Total disability: 5 days Partial disability at 25%: 10 days	Incompetence and negligence of the surgeon
16	Court of Crotone	July 2019	F-42	Goiter	Total thyroidectomy	Bilateral definitive paralysis RLN	Accepted	Biological damage: 16% Total disability: 10 days Partial disability at 50%: 10 days	Lack of post-operation laryngeal nerve integrity check
17	Court of Catanzaro	March 2019	M-N.D.	Goiter	Total thyroidectomy	Right definitive paralysis RLN with aphonia.	Rejected	None	There was no aphonia and temporary paralysis was considered a complication
18	Court of Rome	January 2019	M-67	Goiter	Left thyroidectomy	Right definitive paralysis RLN with dysfagia.	Rejected	None	Complication
19	Court of Rome	March 2018	F-52	Carcinoma	Total thyroidectomy	Bilateral definitive paralysis RLN	Accepted	Biological damage: 40% Total disability: 30 days	Lack of description of the pre-surgical preparation of the laryngeal nerves
20	Court of Naples	March 2018	F-61	Goiter	Left thyroidectomy	Left definitive paralysis RLN	Rejected	None	Complication
21	Court of Catania	February 2018	M-55	Carcinoma	Total thyroidectomy	Bilateral definitive paralysis RLN	Accepted	Biological damage: 10%	Lack of post-operation laryngeal nerve integrity check
22	Court of Rome	February 2018	F-48	Chronic thyroid disease	Total thyroidectomy	Laryngeal edema with respiratory arrest and tracheostomas	Accepted	Biological damage: 40% Total disability: 143 days Partial disability at 75%: 200 days	Failure to prepare laryngeal nerve
23	Court of Sassari	January 2018	M-37	Carcinoma	Total thyroidectomy	Hypocalcemia and anxiety disorder	Accepted	Biological damage: 24% Total disability: 20 days Partial disability at 75%: 25 days Disability at 50%: 25 days	Lack of presurgical recognition of the parathyroid glands
24	Court of Imperia	April 2017	F-25	Goiter	Total thyroidectomy	Dysfagia and aesthetic damage	Accepted	Biological damage: 3% (differential damage from 5% to 2%).	Hypertrophic scar from incorrect suture
25	Court of Naples	April 2017	F-51	Plummer Goiter	Total thyroidectomy	Dysphonia	Accepted	Biological damage: 6–7%	Failure to prepare laryngeal nerve
26	Court of Milan	January 2017	F-73	Goiter	Total thyroidectomy	Dysphonia and dysfagia	Rejected	None	Complication
27	Court of Milan	December 2016	F-51	Carcinoma	Total thyroidectomy	Bilateral definitive paralysis RLN	Accepted	Biological damage: 18% Total disability: 23 days Partial disability at 75%: 90 days Disability at 50%: 120 days Disability at 25%: 120 days	Lack of demonstration of fortuitous event
28	Court of Rome	December 2016	F-23	Goiter	Total thyroidectomy	Aesthetic damage	Rejected	None	Complication
30	Court of Salerno	December 2016	F-N.D.	Goiter	Total thyroidectomy, Arytenoidectomy and Cordotomy	Septic shock and death	Accepted	Damage from death	Incompetence and negligence of the surgeon
31	Court of Milan	September 2016	F-37	Carcinoma	Total thyroidectomy	Dysphonia and dysfagia for bilateral definitive paralysis RLN	Accepted	Biological damage: 3–4% Partial disability at 25%: 250 days	Incomplete informed consent form
32	Court of Livorno	April 2016	F-50	Goiter	Total thyroidectomy	Dysphonia	Accepted	Biological damage: 6% Partial disability at 75%: 30 days Disability at 50%: 45 days Disability at 25%: 45 days	Incompetence of the surgeon
33	Court of Caltanissetta	January 2016	F-52	Goiter	Right thyroidectomy	Dysphonia	Rejected	None	Absence of causal link
34	Court of Monza	January 2016	F-44	Goiter	Total thyroidectomy	Bilateral definitive paralysis RLN	Accepted	Biological damage: 12% Partial disability at 50%: 30 days Disability at 25%: 60 days	Lack of demonstration of fortuitous event
35	Court of Rome	December 2015	F-N.D.	Goiter	Total thyroidectomy	Dysphonia, dyspnoea, hypocalcemia, and reactive depressive disorder	Accepted	Biological damage: 18% Total disability: 60 days Partial disability at 50%: 90 days	Lack of presurgical recognition of the parathyroid glands and failure to prepare laryngeal nerve
36	Court of Naples	November 2015	F-36	Goiter	Total thyroidectomy	Dyspnoea and aphonia for bilateral definitive paralysis RLN	Accepted	Biological damage: 22% Total disability: 9 days Partial disability at 50%: 60 days	Lack of post-operation laryngeal nerve integrity check
37	Court of Milan	October 2015	F-12	Congenital hyperthyroidism and TSH receptor mutation	Total thyroidectomy	Glottic stenosis for bilateral definitive paralysis RLN and hypocalcemia	Accepted	Biological damage: 15% Total disability: 10 days Partial disability at 75%: 30 days Partial disability at 50%: 30 days	Incompetence and negligence of the surgeon
38	Court of Palermo	September 2015	F-N.D.	N.D.	Total thyroidectomy	Hypoparathyroidism	Rejected	None	Prescription of terms
39	Court of Sassari	March 2015	F-N.D.	N.D.	Total thyroidectomy	Dysphonia, dyspnoea, and dysfagia for bilateral definitive paralysis RLN	Accepted	Biological damage: 21% Total disability: 5 days Partial disability at 75%: 30 days Partial disability at 50%: 60 days	Lack of demonstration of fortuitous event
40	Court of Reggio Calabria	February 2015	F-64	Goiter	Total thyroidectomy	Hypocalcemia with tetanic crises	Rejected	None	Complication
41	Court of Milan	January 2015	F-40	Carcinoma	Total thyroidectomy	Surgical suture hemorrhage, respiratory failure tracheostomy done by the surgeon, tracheal injury, post-anoxic encephalopathy, and spastic tetraparesis	Accepted	Biological damage: 90%	Delay in the call of the anesthetist, and tracheostomy performed by the surgeon with tracheal injury
42	Court of Naples	November 2014	F-18	N.D.	Total thyroidectomy	Right definitive paralysis RLN	Accepted	Biological damage: 20% Total disability: 15 days Partial disability at 50%: 30 days	Lack of description of the pre-surgical preparation of the laryngeal nerves
43	Court of Pisa	November 2014	F-32	Thyroid nodule	Total thyroidectomy with MIVAT (Minimally Invasive Video Assisted Thyroidectomy)	Lesion of the right internal carotid artery and thrombosis.	Accepted	Biological damage: 16%	Incompetence and negligence of the surgeon
44	Court of Milan	July 2014	M-52	Carcinoma	Total thyroidectomy	Bilateral definitive paralysis RLN and tracheostomy.	Accepted	Biological damage: 20% Total disability: 30 days Partial disability at 75%: 20 days Partial disability at 50%: 20 days Partial disability at 25%: 20 days	Incompetence and negligence of the surgeon
45	Court of Bari	March 2014	F-63	Goiter	Total thyroidectomy	Right definitive paralysis RLN	Accepted	Biological damage: 9% Partial disability at 50%: 30 days	Lack of post-operation laryngeal nerve integrity check
46	Court of Pisa	April 2013	F-35	Cancer	Total thyroidectomy	Right definitive paralysis RLN	Accepted	Biological damage: 10%	Lack of demonstration of fortuitous event
47	Court of Sassari	January 2022	F-N.D.	Goiter	Total thyroidectomy	Dysphonia, dyspnoea and dysfagia for bilateral definitive paralysis RLN	Accepted	Biological damage: 35% Partial disability at 75%: 60 days Partial disability at 50%: 60 days	Lack of description of the pre-surgical preparation of the laryngeal nerves

## Data Availability

Not applicable.

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
