# Peer review of "Healthcare Liability and Thyroidectomy: When Is the Surgeon Wrong?"

_healthcare, 2023, doi:10.3390/healthcare11040577_

Round 1

Reviewer 1 Report

Please show the strengths and weaknesses of your manuscript 

please also what will be the take home massage of your manuscript

Author Response

strengths: relevance of the topic frequently encountered in medico-legal practice; detailed study of the sentences; understanding of litigation risk prevention measures.

weaknesses: the sentences do not report all the medico-legal data present in the technical consultancy. in some cases it would have been better to read the entire report

take home message: in some cases the healthcare facilities pay for the incompleteness of the medical records even in the absence of a medical error

Reviewer 2 Report

This is an interesting and well conducted article. I have a few comments to improve the paper. 

A flow chart summarizing the selection of 50 sentences, inclusion and exclusion should be included. When using the Pluris Wolters Kluwer database, what were the keywords used to search relevant sentences? 

When searching on the databases, was there any identifiable information that could trace back to the identity of the patient or the people involved? In that case, I believe ethical approval may be needed. Please make sure there are no ethical or legal concerns in Italy.

For any box plot (such as Figure 3) the central horizontal line of the box represents the median. The upper and lower ends of the box represent the upper and lower quartiles, respectively. The two bars represent the two extreme ends, and small dots refer to the outliers. Therefore, in the results section means and SDs were not applicable. Please use medians and interquartile range instead. If the authors prefer means and SDs, use bar graphs or other graphs instead. 

In Table 1, the words “positive” and “negative” were used, while in Figure 2 the words “accepted” and “rejected” were used. Please stick to one set of words for consistency throughout the manuscript. 

The discussion lacks substantial literature on how the complications listed in Table one probably occurred. The recurrent laryngeal nerve is one of the most commonly injured nerves at surgery according to a recent literature search (https://onlinelibrary.wiley.com/doi/abs/10.1002/ca.23696). Anatomically, the awareness of thyroid ima artery is critical when performing thyroid surgery. This variant artery is found in 3.8% individuals according to a recent meta-analysis (https://doi.org/10.1016/j.aanat.2021.151803).

Among the accepted and rejected sentences, did the authors notice any differences in type of surgery, alleged damage, compensation or ground of judgment? This is worthy of discussion.

The word “there” in line 295 should be written as “There”.

The word “13,4” in line 298 should be written as “13.4”.

Font in Table 1 should be changed to a correct MDPI font. 

References should be correctly formatted.

Author Response

we have improved the english, inserted the keywords and the flowchart. we have also included in the discussions the possible complications in the presence of IMA and modified the bibliography. some terms in Italian have been changed to English, we have corrected the formatting of table 1.

we have changed table 3 according to indications

the judgments were without personal data so there is no ethical problem.

Among the accepted and rejected sentences, did the authors notice any differences in type of surgery, alleged damage, compensation or ground of judgment? This is worthy of discussion.

Answer: in the part of the discussions this aspect is extensively covered

Reviewer 3 Report

Although this paper loosely fits the form of a systematic review marginally concerned with thyroid surgery, I feel the topic and presentation would better fit a legal journal or health policy journal, since no new medical/clinical evidence was presented apart from a limited and country-specific insight into one legal system's workings.

Author Response

the reviewer achieved the objective of our study

Round 2

Reviewer 2 Report

The changes are satisfactory. Congratulations. 

Author Response

Thanks for all the revisions that have offered ideas for improving the manuscript

Reviewer 3 Report

I have no objections to the paper itself, but would still feel that it is a better fit for a legal journal, with the discussion pointed in that direction. 

Author Response

(The authors gave the same response as above.)
